# A qualitative exploration of migraine in students attending Irish Universities

**Orla Flynn**[1,2]* , **Catherine Blake**[1,2] , **Brona M. Fullen**[1,2]

**1** School of Public Health, Physiotherapy & Sports Science, University College Dublin, Belfield, Dublin 4, Ireland, **2** Centre for Translational Pain Research, University College Dublin, Belfield, Dublin 4, Ireland

☯ These authors contributed equally to this work.

* orla.flynn.1@ucdconnect.ie

## Abstract

### Introduction

The complex neurological disorder of migraine is prevalent (19%) and burdensome in university students. Qualitative research exploring the lived experience of migraine in students has yet to be conducted.

### Methods

Students clinically diagnosed with migraine were recruited (purposive sampling) from a sample of Irish third-level institutions for a one-time anonymized Zoom focus group or individual interview. Focus group questions were sent to participants in advance. Interviews were iterative. Participants were also invited to submit a drawing. The interviews were audio-recorded, transcribed, and sent to participants for triangulation. Reflexive thematic content analysis was undertaken, data was imported to Microsoft Excel, initial codes were generated, and themes and sub-themes were derived from the codes. The Standards for Reporting Qualitative Studies Checklist (S1 File) ensured study rigour.

### Results

Twenty students from three Irish universities participated (mean age 23.8 years). The four key themes identified were (i) Migraine Characteristics, (ii) Migraine Self-Management, (iii) Migraine Clinical Management, and (iii) Migraine Impacts. Migraine was described as not just a headache but a debilitating sensory experience. A notable high level of self-management satisfaction indicated hopeful coping strategies. However, many participants said medications were ineffective and had side effects, and clinical management could be improved. Additionally, there was a marked academic and social impact of migraine, psychological issues abounded, and several participants worried about finances.

### Conclusions

Migraine is impactful in a cohort of students attending Irish third-level institutions, with students carrying a wide range of debilitating migraine burdens. Students demonstrate an

**Data Availability Statement:** All relevant data are within the manuscript and its Supporting Information files.

**Funding:** The author(s) received no specific funding for this work.

**Competing interests:** The authors have declared that no competing interests exist.

attitude of resilience and determination despite these challenges. Migraine awareness and education campaigns on university campuses are warranted.

## Introduction

Migraine is a complex, genetic, neurobiological disorder [1]. It generally presents as a unilateral pulsatile moderate to severe intensity headache, lasting four to 72 hours, aggravated by routine physical activity. It can be accompanied by at least two of the following: phonophobia or photophobia, nausea, or vomiting. In a third of cases, focal neurological disturbance, experienced as aura may be present [2]. Migraines are prevalent in the general population, with pooled prevalence estimated at 14% [3] and prevalence is higher in young people. A global systematic review and meta-analysis of university students have estimated migraine pooled prevalence rates of 16% [4] and, more recently, 19% [5].

Migraine is among the top ten global causes of disability in people below 50 years of age [6]. Migraine burden and impacts include reduced household and leisure activities, impaired work or school activities, negative relationship impacts, interictal burdens, and financial costs. People suffering from migraine have worse disability or quality of life (QoL) when compared to tension-type headaches (TTH) or non-headache patients, and higher headache frequency has been associated with worse health status [7].

Quantitative research has reported migraine impacts on university students, including college absenteeism, impaired academic performance, social and family difficulties, mental health difficulties, difficulty with activities of daily living (ADLs), and impaired exercise participation [5]. Migraine has also been qualitatively explored in adults, albeit not extensively [8, 9]. Such evaluation has rarely been conducted in Ireland to date, and the student experience of migraine has yet to be qualitatively explored. There has been an evolving recognition of the value of utilising qualitative research methodologies to deeply explore the patient perspective, which can be difficult or impossible using other research methods [10]. This is particularly relevant in those living with chronic pain conditions where there is an underrepresentation of qualitative literature, and only the person in pain can report changes in their life quality [11].

Thus, qualitative research is warranted to enhance understanding of diverse conditions such as migraine. Research in discrete populations such as university students is valuable, as smaller populations are likely to differ from the general population in some respects. For example, they may be healthier, wealthier, poorer, and more or less educated [3]. Additionally, as migraine is so prevalent in younger people, it is crucial to understand this condition in a cohort where it is highly disruptive. Further, as university students progress from their studies into the workforce, studies of this population have broader relevance. Hence, the current study aimed to establish the migraine experience in an Irish university student population via one-time anonymised online individual or focus group interviews. This study serves to address the qualitative knowledge gap regarding the experience of migraine in young people.

## Methods

### Ethical approval

Ethical approval was obtained from the University College Dublin (UCD) Human Research Ethics Committee (HREC): Reference number: Flynn-Fullen-18-83.

## Study advertisement

The study was advertised on the UCD campus via poster advertisement (S1 File), word of mouth, and various social media channels. Due to data protection regulations, students could not be directly targeted via email. Within UCD, the UCD Students Union, UCD Student Health and various sports clubs agreed to share the posters. Social media graphics were disseminated via LinkedIn, Facebook, Instagram and Twitter.

Recruitment from other universities was conducted by directly contacting the communications departments or reception desks of all major Irish third-level institutions. The Irish Society for Chartered Physiotherapists (ISCP), the Irish Universities Association and the Migraine Association of Ireland (MAI) were also contacted. After potentially interested participants emailed the Primary Investigator (PI), the Participant Information Leaflet (S1 File), a consent form (S1 File), a demographics questionnaire (Table 1), a sample illustration of a migraine experience (S1 File) and the list of focus group questions (S1 File) were provided.

## Inclusion criteria

Students who self-reported having been clinically diagnosed with migraine by their General Practitioner or neurologist in the demographics questionnaire. No physical or neurological examination was conducted.

**Table 1. Demographics questionnaire.**

| Questions | Answers |
|---|---|
| 1. Have you been clinically diagnosed with migraine by a medical professional (e.g., General Practitioner, Neurologist) | Yes/No |
| 2. What is your age? | Open-ended response |
| 3. What age were you when your migraines began? | Open-ended response |
| 4. What is your gender? | Open-ended response |
| 5. What university are you attending, and what course are you studying? | Open-ended response |
| 6. What type of migraines do you have? Tick all that apply. | Migraine without aura<br>Migraine with aura<br>Menstrual migraine<br>Vestibular migraine<br>Abdominal migraine<br>Hemiplegic migraine<br>Retinal (ocular) migraine<br>Refractory (difficult to treat/does not respond to treatment) migraine<br>Other/Unsure |
| 7. How often do you get migraines? | Episodic (less than 15 days per month)<br>Chronic (greater than 15 days per month) |
| 8. Is there a hormonal pattern to your migraines? For example—do they come on at certain times of the month (if female)? | Yes/No/Unsure |
| 9. Is there a circadian pattern to your migraines? For example, do they come in the morning, afternoon, or evening? | Yes/No/Unsure |
| 10. If you take migraine medications, how often? | Daily or almost every day of the month<br>Greater than twice per week<br>Less than 15 days per month<br>Greater than 15 days per month<br>Rarely<br>Never |
| 11. Do you think your knowledge about migraines is? | Poor/Adequate /Good |
| 12. Do you keep a headache diary? | Yes/No |

### Exclusion criteria

People who were not registered to a university, school students, or university students without a clinical migraine diagnosis.

### Informed consent

Potential participants provided informed consent in advance of the interviews. If participants decided to proceed, they completed the written Informed Consent form (S1 File) via an electronic link. Participants could choose to withdraw from the study at any time, including after the interviews concluded and were analysed.

### Sample size

A purposive sample of undergraduate and postgraduate university students who attended any Irish third-level institution was invited to participate. Given the challenges and limitations of conducting research during COVID-19, the research team reached a consensus to offer potential participants both focus groups and individual interviews.

Health science and migraine literature was used to inform the sample size decision [8, 12, 13]. For focus group saturation, greater than 80% of all themes are discoverable within 2–3 and 90% discoverable within 3–6 focus groups [14]. Similarly, qualitative interviews can reach saturation at relatively small sample sizes (9–17 interviews) [15]. Collectively, an adequate average sample size for generic qualitative studies is 20 participants [16].

However, it is essential to note information power rather than data saturation [17]. This concept shifts the focus from sample size to data collection quality. Interviews should be iterative, and longer interview lengths may generate richer information than a larger sample size. Six in-depth interviews with open-ended questions lasting 60 minutes or more will likely yield richer data than twenty 10-minute interviews that elicit only surface-level responses [18]. Thus, interview quality and length matter more than quantity.

Further, researchers using reflexive thematic content analysis (TCA) should note that judgments about data quantity and when to stop data collection are subjective and cannot be wholly determined before data analysis [19]. Thus, based on the factors above and perusal of additional qualitative study literature [20, 21], an a-priori sample size of 15–20 participants was selected, allowing for 10% attrition rate. Reflexive analysis was used throughout the data collection process to assess whether to cease data collection at 20 participants or if further data collection was needed.

### Timeframe

The focus group times were organised via an anonymised Doodle poll sent to potential focus group participants, and individual interviews were conducted between the primary investigator and the students. Focus group size depended on which times suited which students to attend. Study advertisement and data collection began on December 6th, 2021, and concluded on May 31st, 2023.

### Language

The interviews were conducted in English.

### Research design

A qualitative research design was used. Quantitative data was collected for the demographics portion of the study.

**Quantitative data research design.** Quantitative data was collected via an electronic, anonymised participant demographic questionnaire (Table 1), administered via the secure online platform Qualtrics [22], which sought to capture (i) participant demographics, (ii) migraine history, (iii) migraine profile (type, triggers, pattern), (iv) migraine management (medical and self-management strategies) and (v) self-rating of migraine knowledge. Participant demographics included age, gender, university attended, undergraduate or postgraduate stage, migraine type and usage of a headache diary (Table 1). An optional migraine illustration depicting participants' migraine experience could also be provided by participants using the drawing function in Microsoft Word (S1 File).

**Qualitative data research design.** The study was located within an interpretivist research paradigm [16], where knowledge is co-constructed by the researcher, the research participants, and the collaborative research team [23, 24]. The motivation for adopting this methodological approach is to discover meaning and understand the participants' experiences contextually [25]. The Standards for Reporting Qualitative Research Checklist [26] was followed to facilitate high-quality reporting (S1 File).

The interpretivist descriptive approach is constituted by constructivist epistemological philosophy, meaning that knowledge is not absolute but is socially constructed through the subjective person who experiences it [27]. This approach aligns with the study's aim to understand participants' subjective reality about their migraine experience. After the questionnaire (and optional illustration) were completed and returned, a Doodle poll (using a 'hidden feature') was used to arrange suitable times for participation in the audio-recorded Zoom focus group or individual interview.

When scheduling the focus group, the 'hidden' feature on Doodle Poll was used so that only the lead researcher could access participants' names. Participants were also given a code they used in place of their names and instructed to have their cameras switched off during the interviews. The audio-recorded interviews were conducted by members of the research team (BMF, OF) using the Zoom platform [28] and scheduled for approximately 60 minutes. These interviews were used to generate an in-depth understanding of the migraine experience in a student population, using a battery of open-ended questions (S1 File) informed by the qualitative literature [8, 12, 13].

The study was also iterative—further questions were generated based on participant answers—which added to the depth of information collected. In a qualitative study, it is possible to increase knowledge of the sufferers' perspective and gain an in-depth understanding of a specific phenomenon. The method aims to make sense of or interpret phenomena regarding the meanings people attribute to them [29]. Using this methodology allows for more decadent data collection to explore the experience of students diagnosed with migraine.

## Data management and analysis

**Quantitative data analysis.** Descriptive statistics were used to report the Demographics Questionnaire and compiled using Microsoft Excel.

**Qualitative data analysis.** Following the recommendation of Miles and Huberman (1994) [30], all focus groups and interviews were audio-recorded. The recordings were transcribed immediately using the transcription function in Zoom, and the Zoom recordings were used to cross-reference the automatic transcription to account for vernacular language. The completed transcripts, which were sent to all study participants for triangulation, formed the basis of the database. Participants had one week to return the transcripts. Each participant was identified by a unique number (1–20), and the transcripts were line-numbered to allow for data coding and citations, e.g., Participant 1, Ln1.

The data was rigorously analysed using reflexive thematic content analysis (TCA) [25, 31], which aims to attain a condensed yet broad description of phenomena [32]. This approach is about the researcher's subjectivity as an analytic resource and their reflexive engagement with theory, data, and interpretation rather than measures of inter-coder agreement [33]. The analysis was primarily conducted by the lead researcher (OF), who meticulously ensured the robustness of the findings. The data was inductively analysed, thus developing themes rather than using a preconceived list [34].

The reflexive thematic analysis, a rigorous and systematic approach, involved six distinct steps [25, 31]. The steps involved data familiarisation, the development of a coding scheme, the application of second-level coding, code review, the creation of themes, and the finalisation of themes and subthemes.

In step one, OF achieved data familiarisation through repeated listening of the interviews alongside reading and re-reading each interview transcript several times. This thorough data immersion ensured a comprehensive understanding of its meaning. In step two, a coding scheme was developed via Microsoft Excel (S2 File) using the preliminary conceptual framework to facilitate data reduction. Key statements were grouped into categories (OF) (S2 File), generating various potential data interpretations. In step three, OF applied second-level coding (S2 File), allowing for significant themes to be identified (S2 File). Minor themes were also identified where relevant (S2 File). All themes were reviewed and refined in step four.

Step five involved the research team (OF & BMF) collaboratively reviewing a sample of transcripts, following which they discussed and agreed on the final themes and subthemes. Due to information repetition, it was agreed that no further study participants needed to be recruited. The final themes and subthemes were refined and imported into a codebook (S2 File). Quotes were provided for each theme (S2 File). Step six involved using the codebook to write a coherent 'story' about the data.

**Research team, reflexivity, and trustworthiness.**   Reflexivity is essential to promoting trustworthy and quality results [33, 35]. The first author (OF) is a part-time PhD candidate, physiotherapist, and personal trainer with experience in health and exercise clinical practice. The other research team members are two academic physiotherapists (BMF & CB) with qualitative research experience.

Researchers should convey not only why a dataset is sufficient but also how data were interpreted and what they contribute [16]. Thus, the coding spreadsheet (S2 File), which was used, allows the study to be transparent and reproducible. Each participant's example quotes and comments (S2 File) were imported into the spreadsheet to allow for data extraction ease and rapid cross-checking of information when writing the manuscript.

Reflexivity memos (S2 File) were also completed throughout the analysis process (OF) to engage with and identify possible subjective biases brought to the analysis [35]. This reflexive process, discussed with the other research team members (BMF & CB), plays a crucial role in enhancing the trustworthiness of the work by ensuring the researchers' awareness and management of their own biases.

Additionally, an essential feature of qualitative research is data validation [36]. Thus, triangulating the transcripts added study rigour. Approaching the data extraction and analysis in this methodological manner demonstrates the results' credibility, transferability and dependability.

## Results

Twenty students from three Irish third-level institutions participated in this project: University College Dublin (UCD) (N = 17), Trinity College Dublin (TCD) (N = 2) and University of

**Table 2. Participant demographics.**

| (N = 20, %) | N, % | N, % |
|---|---|---|
| **Age, mean (range)** | **Gender** | *__Migraine type__ |
| 23.8 (19–46) | Female (15, 75%) | Aura (14) |
| **Age onset, mean (range)** | Male (5, 25%) | Without Aura (8) |
| 13.5 (6–23) | | Retinal (7) |
| | | Vestibular (6) |
| | | Menstrual (4) |
| | | Refractory (2) |
| | | Hemiplegic (1) |
| | | Abdominal (1) |
| | | Other/Unsure (2) |
| **University & stage** | **__Migraine frequency** | **__Knowledge rating** |
| UCD (17, 85%) | Episodic (14, 80%) | Good (6, 33.5%) |
| TCD (2, 10%) | Chronic (4, 20%) | Adequate (11, 61%) |
| Ulster (1, 5%) | | Poor (1, 5.5) |
| **__Patterns** | **__Medication frequency** | **__Headache diary** |
| **Hormonal** | Daily/>15d/m (7, 39%) | Yes (8, 44%) |
| Yes (5, 28%) | Rarely/Never/<15d/m (11, 61%) | |
| **Circadian** | | |
| Yes (5, 28%) | | |

Footnotes: UCD = University College Dublin, TCD = Trinity College Dublin, UG = undergraduate,

PG = postgraduate, PG MSC = Masters, PG PhD = PhD, >15d/m = greater than 15 days per month, <15d/m = less than 15 days per month.

*Several participants reported multiple migraine types therefore % = n/a.

**Two participants did not fully complete the demographics, thus, percentages were calculated from 18 people from migraine frequency onwards.

Ulster (UU) (N = 1) (Table 2). Seven individual interviews and four focus groups were conducted. The focus groups consisted of two focus groups with four participants each, one group with three participants and one group with two participants. The focus groups and interviews ranged between 60–90 minutes in length. The students' mean age (range) was 23.8 (19–46). Most of the participants were female (N = 15, 75%), and a large number reported migraines with aura (MWA) (N = 14, 70%). Migraine frequency was predominately episodic, although almost 20% of participants (N = 4) reported chronic migraine. Nearly forty percent of participants (N = 7, 39%) reported taking daily or almost daily medication. A third of participants (N = 6, 33.5%) reported being confident of their migraine knowledge (Table 2).

## Themes

Four key themes emerged from the interviews (Table 3). The results are presented in detail under the four main (and associated) sub-themes illustrated with examples from the transcripts.

## Migraine characteristics

This theme comprised two subthemes: (i) profile and (ii) triggers.

**(i) Profile.** Migraine profile consisted of descriptions of the distinct attack stages. Almost half the participants described migraine as not just a headache but a debilitating experience. A harrowing depiction provided by one participant illustrates this well (Fig 1). All but one student described prodromal disturbances, and all participants described typical migraine symptoms of severe throbbing headache, phonophobia, photophobia, nausea, vomiting, and movement-related symptom aggravation. Many participants described additional attack

**Table 3. Overarching themes and subthemes.**

| Overarching themes | Migraine characteristics | Migraine self-management | Migraine clinical management | Migraine impacts |
|---|---|---|---|---|
| Subthemes | (i) Profile<br>(ii) Triggers | (i) Strategies<br>(ii) Barriers & facilitators | (i) Medications<br>(ii) Clinical pathway | (i) Academic<br>(ii) Social<br>(iii) Psychological |

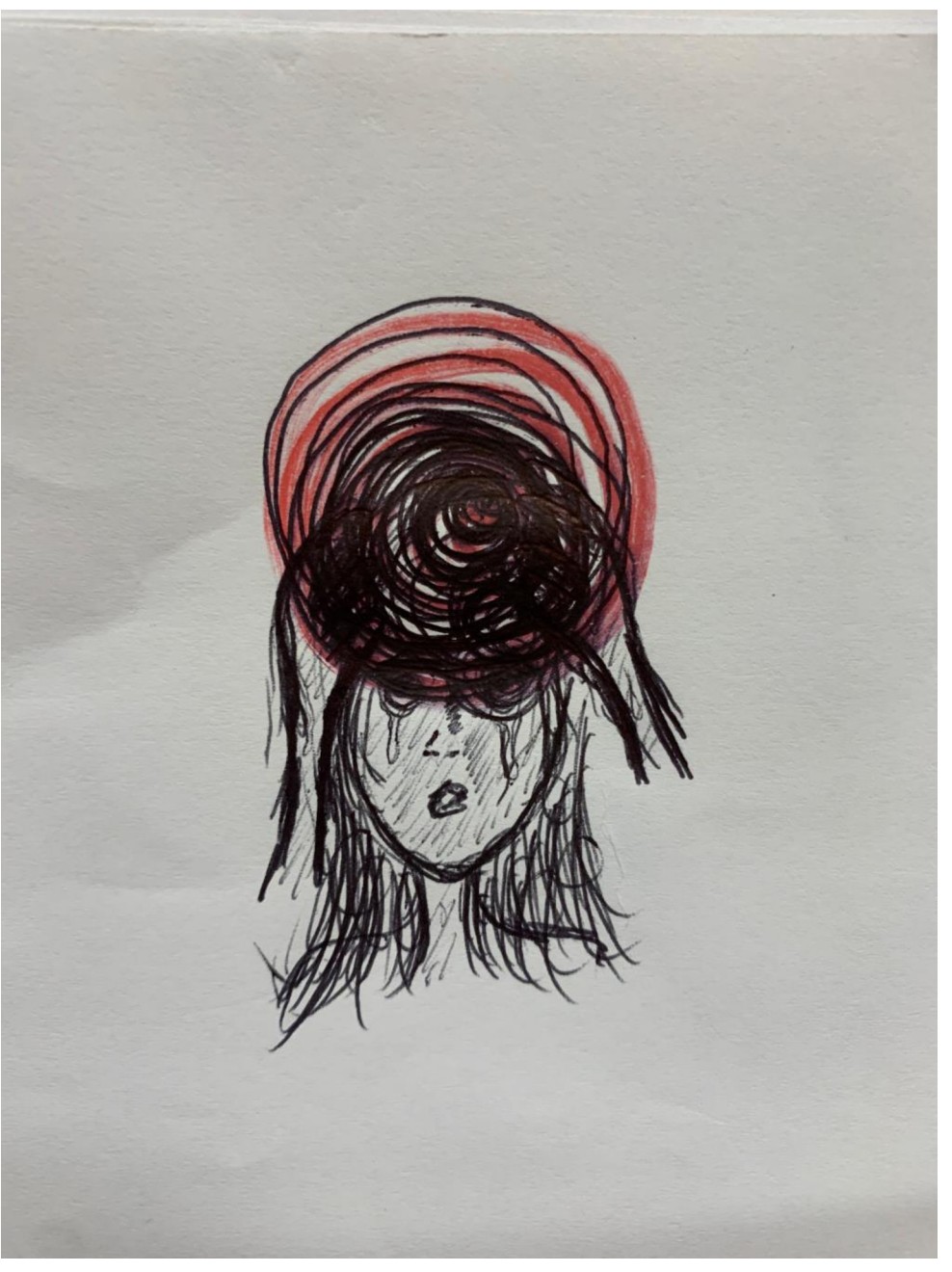

**Fig 1. The migraine experience.**

symptoms such as cognitive dysfunction. Almost all participants described transient focal neurological disturbances of aura, such as visual disturbances, paresthesia, aphasias, vestibular dysfunction, cognitive dysfunction, mood changes, and fatigue. Distracting postdrome symptoms of fatigue, brain fog, and difficulty functioning were experienced by almost half of the participants.

"....blurred vision....volume being turned up....pins and needles....my tongue....does not feel like my own hands....difficulty finding words....agony....extreme distress....such intense pressure building up in my head, that the only natural thing to do it to compress it with my hands." P6, Ln20

".... my speech becomes very confused and then it's the same with numbness down one side of the body and issues with my vision....like pain and in my temples or behind my eyes.... I'll get quite dizzy or I'll feel very faint as well." P4, Ln43

**(ii) Triggers.** Participants were subject to migraine triggers in many domains unavoidable in a university environment: behavioural (e.g., disrupted sleep or episodes of stress), dietary (e.g., aversion to monosodium glutamate and processed sweets), environmental (e.g., visual stimuli or weather changes), and physiological (e.g., hormonal factors or intense physical activity).

"....if I don't get enough sleep, or if I have a full day in college I almost always get one." P1, Ln134

"Preservatives....and any type of sweetener so like saccharine,

aspartame, sucralose....so I can't have any diet sodas, any sugar-free

sweets...MSG also triggers it as well." P20, Ln4

## Migraine self-management

This theme consisted of two subthemes: (i) strategies and (ii) barriers and facilitators.

**(i) Strategies.** All participants engaged in self-management strategies, which was their preferred approach. The predominant strategy was sleep, followed by healthy lifestyle changes and trigger avoidance. Self-management also required a very structured daily routine.

"started sleeping early and waking up early" P7, Ln64

"....kept a log of everything I ate....trial and error method....lifestyle is a huge factor.... weight-training.... Meditation." P7, Ln64

Holistic treatments such as taking magnesium and using lavender were used to facilitate stress management by a small number of participants.

"I like lavender....essential oil....just breathing....I know it's really counterproductive, but just like putting on a movie....distracting my brain...." P17, Ln111

**(ii) Barriers & facilitators.** Management barriers and facilitators were generally conflicting. A quarter of participants said it was not always feasible to self-manage during an attack when having to proceed with responsibilities.

"Um, which can be challenging if it happens when I'm in college, because um you know I commute so it's it's a good bit away. So sometimes you kind of have to either just stay put or run home, and just to kinda get over the worst of it" P9, Ln226

"if I have the luxury of being at home when I develop a migraine, it's usually into my bedroom, curtains closed, lights off." P14, Ln37

Nearly 75% of participants said external barriers such as others' misunderstanding of the condition created management barriers or felt that their faculty or workplace would not support migraine as a disability, and they may be penalised for having the condition.

"The biggest issue is trying to get people, especially lecturers, to understand that a migraine is not just a headache. . .. it's something much worse than that. . ..I'm allowed x y z allowance, and some of them very begrudgingly give it to me. They don't see migraine as an excuse. . ..I wish you could experience what I experience for one day. . .. do the public understand what migraine is?" P15, Ln333

"College isn't great at accommodating people with conditions such as migraine so often I have to go to go in. . ..have to make up the time and do extra papers as punishment for taking time off for being sick." P17, Ln111

". . ..I explained like, you know, I really am trying my best, but sometimes I can't make it into the office and that's not because, like I'm lazy or I don't wanna go. It's because, like, I actually cannot get there" P20, Ln42

Although the universities did provide concessions which almost half of the participants utilised, some of these participants reported difficulty accessing on-campus accommodations.

"Yeah, they gave me a room with the code. . ... library. um, I'm around the area. So that's not too bad, but it's more like you have to book it, like I have to in the moment, book it online, and then find my way there and put in the passcode. And I'm not 100% sure that's something that I'd be able to actually find my way around on my own without any help" P13, Ln127

Perception of migraine as a disability was conflicting as over a quarter percent of participants exhibited internalised stigma, either minimising their condition or exhibiting an imposter syndrome, viewing migraine as a disability in others but not themselves.

"For my state exams and my college exams I had disability accommodations because of the migraine. But sometimes you kind of feel like I need them, but I also, it's not the same as other disabilities. So you kind of feel a bit like an impostor as well, you know. And so yeah, it's still in that grey area." P9, Ln397

"it seems to be a general view that people who are looking in from the outside. I myself refer to my migraines as headaches. I don't know if that's just because I like to minimise it to people around me, so they don't worry" P1, Ln265

Migraine management facilitators included family and friends as support networks. Encouragingly, half the participants reported external facilitators, saying that their family and friends were supportive, particularly in families where other family members suffered from migraine.

I'm very lucky because a lot of my family would experience migraines as well, so it's kind of you know, you know how to deal with it I suppose. . .." P6, Ln192

Internal facilitators were also present whereby half the participants were either registered with disability services or felt that the university or workplace supported their condition, once they were made aware of the migraine diagnosis, though the concessions made by the university were minor in nature.

"I'm registered with the Disabilities office, so I guess, like an extra ten minutes for every hour, which still help during the early morning exams" P15, Ln154

Although their attitudes polarised between positive and negative, half the participants demonstrated positive facets of illness acceptance and condition management accountability, striving to proceed with daily life as normally as possible.

". . .. I'm of the mindset, you know, like death and taxes, migraines are just around." P5, Ln124

"I think I kind of came to like accepted that maybe it isn't going to be managed." P6, Ln252

"I'm really proud of myself for how much I've done with the migraines, you know,

and how hard I've just worked and powered through to get where I am" P20, Ln50

### Migraine clinical management

This theme consisted of two subthemes: (i) medications and (ii) clinical pathway.

**(i) Medications.** Most participants reported taking over-the-counter or prescription medications for migraine, ranging from simple analgesics to triptans to beta blockers. No participants mentioned novel migraine treatments such as calcitonin-gene-related peptides (CGRPs). Over three-quarters of participants reported barriers such as medications being ineffective or partially effective but not able to halt attacks.

"I do believe that these have reduced the severity. . ..but I have never been able to prevent them. . ..also limited my choice of hormonal contraceptives. . ..which resulted in the long-term use of different antibiotics to treat my acne which brought its other problems." P6, Ln20

Many of these participants would prefer not to take medications due to side effects.

"Over-the-counter medicines did not help me because my migraine was too strong for that, so I have always taken prescribed medication for my migraines. And yes, I gained about 15 kilos at least, like 15 to 18 kilos, over the course of 3 years" P7, Ln114

**(ii) Clinical pathway.** Participants obtained knowledge primarily from clinicians and the Internet. The overall preference was verifiable information, or they would be cautious or dubious about the information received. Students were limited in the scope of migraine pathophysiology knowledge, with many choosing not to answer this question.

"I would have read one or two clinical articles. . .. my diagnosis came from my GP, but to learn a bit more, just through Google Scholar." P4, Ln14

"Yeah, I'd be similar to that, the Migraine Association. . . .they share a lot of good information." P8, Ln28

Many participants said their ideal consult would be leaving with more information about their migraine, but several had yet to be provided with information by their clinician.

". . . .um doctors and clinicians haven't been as helpful in my experience." P8, Ln28

Just under half the participants said they felt partially or fully listened to by their clinician, and feeling listened to would be their preferred consultation strategy.

". . . .I don't expect the GP to have this magical cure for me. So I suppose, just being, like a feeling that you have been listened to I suppose." P6, Ln142

Many participants had mixed opinions about their clinical experience, with a small number having negative experiences and over half of participants having conflicting clinical experiences. General practitioners are reportedly well-meaning but limited in migraine expertise, with some oversights with pharmaceutical management.

"she's like, oh yeah, that's a migraine. And kinda just left it at that. . . .after that I was working in the pharmacy. I didn't even know that there were things like prescriptions. . . .I was already on the contraceptive pill. . . .over two years later. . . .I was with another doctor. . . .'oh, like when you have migraines like it's not recommended that you take the combined pill'" P18, Ln34

"Yeah, and like, for a couple of years I was on estrogen birth control and I didn't realize it. And then someone said, you can have like a stroke if you're on this. And with the migraines, and I had like no idea" P20, Ln12

". . . .listening to doctors, I don't think that's always. . . . its not what you want it to be. . . ." P11, Ln16

Thus, almost half of the participants' general practitioner (GP) attendance was irregular, generally for re-prescription only.

"Similarly, um, I had one appointment for the initial diagnosis, and then I think maybe one follow up? And yeah, it's self-management after that." P10, Ln47

Over a quarter of participants believed neurologist care is or would be effective. However, they needed help accessing this, and several participants were on neurologist waiting lists, so they were not currently engaging with specialists.

"Feel like there's not really a solution until I get to see a neurologist but I'm just on a waiting list, so I feel like I'm in a bit of a limbo." P3, Ln417

Overall, almost half of the participants were satisfied with migraine management. Several participants were confident with self-management rather than medical management, which was conflicting.

"....now they've kind of come back with a vengeance.....I'm not dissatisfied. But yeah, I think there's more that could be uh going better..... So that feeds into that whole idea of the waiting lists...." P9, Ln165

"I'm happy not with the way my migraine is managed, but I'm happy with the way I manage my migraine um.....with some help from medical profession." P12, Ln348

Encouragingly, several participants also reported a positive medical experience.

"....I am also on antidepressants, and but that's I like, started them like two or three years after I was experiencing the really terrible migraines.....an SSRI which is a called Lexapro. But I think that it really really helps as well with like. Obviously, my like anxiety levels are much more balanced, and I think that it has reduced the like probability of getting or like experiencing migraine symptoms as a result...." P16, Ln256

I think, like the current neurologist, I have now. She really knows what's going on. I feel really confident in her ability to diagnose me and help kinda get my life back on track and kinda get my life back. So I feel really good about her" P20, Ln 40

## Migraine impacts

This theme consisted of three subthemes: (i) academic, (ii) social and (iii) psychological impacts.

**(i) Academic.** Participants used powerful and emotive language to describe migraine impact, similar to other qualitative research [8, 37]. Three-quarters of participants said there was an academic impact—absenteeism, presenteeism, concentration issues, or difficulties getting assignments in on time.

"So I've missed numerous college exams.....even social events and shifts in work...." P6, Ln163

"Sometimes I can't make it into the office....not because I'm lazy....like, I actually cannot get there." P20, Ln42

Daily life included over half of the participants having time loss per day managing migraine.

"And yeah, I I'd say it would take time out of my day to a certain extent yeah." P9, Ln199

"lose nearly the first couple of hours of being awake with just drowsiness, just trying to kind of push the rest of this medication out of my system." P15, Ln154

A few participants stated that commuting was difficult or dangerous during an attack.

"and also, my biggest concern is driving, because since I have a migraine with aura, and I cannot properly see, it can be very dangerous, especially at night...." P19, Ln113

"....I remember having to sleep under my motorbike for half an hour on the motorway on the hard shoulder, because I couldn't see a thing.....nowhere else to go.....endanger yourself...." P12, Ln425

**(ii) Social.** Relationships, socialising, and exercise capabilities were affected for almost all participants. Migraine affected these pursuits predominantly due to an inability to drink alcohol alongside medications, difficulty staying out late, or committing to planned activities.

"Social life, it can affect that hugely. . ..I found alcohol. . ..you have like a super hangover. . . . And yeah I suppose it can be hard to keep connections with people if you keep cancelling on them. . .." P2, Ln294

"if you don't drink, they're like what's wrong with you." P17, Ln227

"it can kind of make you feel a bit outcast sometimes if they're saying, oh, you know you just have a headache, I had a headache the other day." P1, Ln265

**(iii) Psychological.** Almost three-quarters of participants had psychological issues such as mood disturbances and interictal anxiety.

"I'm always worried coming up to an exam season. . ..I felt I was really down. . ..just really negatively affected my mood" P4, Ln253

Worrying about future work prospects also contributed to anxiety in a small number of participants.

"And just trying to figure out kind of like, you know, how sustainable, is working in a really high-pressure industry. I think that's also kind of been on my mind a lot, too" P20, Ln46

Almost half of the participants demonstrated anxiety about the financial impacts of affording medications or accessing specialist care.

"The finances have been cause of a kind of a point of anxiety for me I'm very lucky in that my parents still pay for my medication and my health expenses. . ..there's been times, where I will not take a tablet because I'm aware that this single tablet cost me six euro. . ..very expensive, particularly the triptans. . ..then on top of GP visits. . ..they're extremely expensive. . .." P1, Ln406

"Yeah I would say the same, especially what resonated with me was the rationing thing, like on a bad month, or if I don't get many shifts in work or anything I will just kind of have to deal with it. . ..because I know I can't really afford more medication." P3, Ln412

The overarching theme from three-quarters of the participants was a resounding perception of freedom for the participants if they did not have migraine. A quarter of participants said they would be excited to see what they could have accomplished without migraines.

"It would be a sense of relief. I think it would boost my confidence, and I would be really excited to see what I could accomplish. . ... there's also like, I'd be interested to see if, like I feel like I'm fighting with one hand behind my back. If I was to get the other hand out, like, what I would be able to accomplish is something that like I really, I really wanna see that." P20, Ln50

". . . .would allow me to I suppose realise my full potential. . ..I would have loved to have seen, what I would have been, kind of capable of, even in terms of Leaving Cert points and stuff, had I not had migraine, had I not missed the guts of 50 days a year in school. Which is always going to have a huge impact." P2, Ln344

## Discussion

Migraine is far more than just a headache. The overarching theme that emerged for migraine characteristics was that migraine was not just a headache but a debilitating sensory experience. Although three-quarters of students reported episodic migraine in the demographics form, and aside from the debilitating classical and focal neurological symptoms of the attack phase, almost all participants reported prodrome and nearly half reported postdrome. This resulted in the migraine experience spanning days rather than hours and being highly disruptive. Though patient perceived migraine triggers may be early attack manifestations [38], all participants in the current study described triggers in every domain, likely due to migraine triggers being widespread in the university environment [5]. This led to students having difficulty avoiding migraine whilst proceeding with their courses.

The principles of SEEDS (Sleep, Exercise, Eating, Headache Diary, Stress Management) encompass a useful self-management rubric [39]. Students tended to follow this, particularly the sleep pillar, and sleep has well-established therapeutic benefits [40, 41]. The SEEDs principles were not explicitly stated by the students, and this behaviour appeared to be experientially driven rather than derived from clinical education.

A university sample is, by definition, highly educated, and migraine knowledge deficits in this sample are likely to be accentuated in other groups [42]. Many students chose not to answer the question on migraine knowledge or needed to be more specific about migraine pathophysiology. Migraine pathophysiology literacy in the current study was low among participants. Since pathophysiology has changed over time [43, 44], patients may expect practitioners to provide novel information. However, in the current study, the participants felt their practitioners needed to educate them sufficiently, and thus, they aimed to educate themselves. It is encouraging that the students sought verifiable management knowledge as non-governmental organisations have demonstrated the overall highest quality migraine management information, compared with Googling and YouTube, which provide information of varying quality [45, 46].

Regarding pharmaceutical management, in recent qualitative research, participants consistently ranked pain relief/pain absence as their highest acute pharmaceutical treatment priority. Additional priorities included limited side effects and reliable medication efficacy [47]. The current study did not rank order preferences but echoes Mangrums' findings, particularly the students' aversion to medication side effects. The students' conflicting attitudes towards medication may have affected their perception of their primary care provider, as GP prescriptions of medications that patients believe might have side effects have been negatively correlated with patient trust [48].

In the current study, many participants preferred not to take medicines, were dissatisfied with medication efficacy, and reported side effects. Although OnabotulinumtoxinA (Botox) is an evidence-based medical practice [49], only two participants mentioned it. Clinician attendance was also sporadic, and clinical experiences conflicted. Several factors can lead to underutilising appropriate medical treatments, including poor understanding of treatments [42]. The student's low migraine literacy was also reflected in their limited knowledge of novel migraine treatments, as no student mentioned being informed about emerging treatments. Knowledge acquisition is not solely incumbent on the patient; rather it is the duty of the clinician, and this further speaks to the negative and conflicting clinical experiences discussed within the interviews.

Patient clinician trust also relates to provider satisfaction and patients learning about their medical conditions [50]. This idea is supported by some participants in the current study reporting positive neurologist experiences, though many expressed waiting list dissatisfaction.

Migraine is one of the top reasons for patients seeking neurology services, but waiting lists are lengthy [51]. Globally, new patient waiting lists for specialised headache centres can extend to 14 months [52]. There is an ongoing shortage of consultant neurologists in Ireland [53, 54], resulting in prolonged hospital waiting lists [55]. Overall, participants expressed some self-management satisfaction but conflicting medical management satisfaction.

People living with migraines must employ disease acceptance [8, 9, 12]. Encouragingly, half the participants in the current study utilised external facilitators for migraine management. Students disclosing (or registering) their medical condition with their university might result in more favourable accommodations [56]. However, the academic and disability accommodations provided to participants in the current study were sparse and not necessarily reflective of real-world considerations a student with migraine may require. Some of these students were unaware of available quiet spaces on campus for those with illnesses. Of those students who were, they outlined the detailed steps needed to access these facilities, such as inputting a code into a code pad. These strategies may be impractical, particularly for the three-quarters of students who reported visual and motor disturbances of aura.

Additionally, almost half of the participants demonstrated a complete lack of knowledge of university academic and disability accommodations for migraine. There was also a sense that university faculties could be more supportive. University faculty education about the impacts of illnesses and how to frame empathic responses to student disclosures are critical for those students with individual chronic illnesses [56]. Further, workplace migraine management education has been associated with an absenteeism reduction and productivity increase [57], and ergonomic considerations could also be helpful [58, 59]. These practices could easily be utilised in a university environment.

However, while ostensibly aimed at equity, accommodation practices often reinforce individualised normative functioning standards, and rather than making systemic environmental changes, individual personal adjustments are made [60]. Tellingly, in the current study, all participants have made lifestyle adjustments. This paradigm failure is framed as an individual problem; thus, those with migraines will internalise their inability to work as a personal rather than a social failing [61]. Migraine is challenging to destigmatise, and the social environment impacts the variable degrees of disability [61]. Thus, students can be reluctant to disclose migraine [61], perhaps owing to this potential stigmatization. Ensuingly, over a quarter of participants minimised the condition to others or did not consider migraine a disability. Additionally, in the general population literature, participants yearned to share experiences to understand and contextualise their condition [8]. Reflecting this, a few participants in the current study said they would like to be part of a support group as they felt isolated living with this condition.

The impacts of migraine on professional, private, or social domains and daily activity limitations have been demonstrated throughout all migraine phases in both quantitative and qualitative research [8, 9, 62]. In the current study, migraine greatly impacted every domain. Almost all participants said there was a marked academic impact. In the general population, sufferers also report migraines impacting their work and daily activities [8, 63]. In the current study, although they had not yet transitioned from university, a few participants worried about future workplace difficulties, suggesting they believed their condition would still not be well managed by the time they left university.

People with migraine report feeling misunderstood and isolated in relationships, as friends and acquaintances can struggle with legitimising migraine[8,] and the students in the current study echoed this. College is a time when individuals value making and fostering interpersonal connections [64], and university students are already at higher risk of feeling lonely than other

population groups [65]. Thus, it is sobering that students would be psychologically struggling when they should be in the prime of their lives.

Emotional impacts of migraine are not solely during the migraine attack phase but also during the interictal period [9]. The current study reflected these findings as students spent much of their time struggling with the different challenges migraine imbues. Psychological distress is common among people with migraine, who suffer a lot and can become overwhelmed [8]. In the current study, the overall impression of not having migraine anymore would be the freedom for the students to live life in an unconstrained and unlimited way.

## Study strengths and limitations

Before this study, qualitative research on migraine in young people had yet to be conducted in Ireland. This novel study encompasses data from a sample of Irish third-level institutions and undergraduate and postgraduate student populations. It provides insightful data on the migraine experience, highlighting the management gaps and the myriad of impacts that students experience. This study was thoughtfully designed and conducted through a mixture of individual interviews and focus groups, reflexive thematic analysis, sufficient interview lengths and a rigorous, transparent and reproducible data analysis process. Data collection was concluded when data sufficiency was achieved via information repetition. The iterative process of the study added further depth and rigour. A limitation is that the study occurred during the COVID-19 pandemic, which affected participant recruitment despite widespread networking with universities and professional organisations. However, recruitment issues were adjusted by facilitating both individual and focus group interviews. Most of the study participants were female, which may limit the generalizability of the results. No physical or neurological examination was conducted; thus, secondary headache exclusion was impossible. The sample size was small, however appropriate due to the aforementioned sample size factors, and within appropriate and standard ranges for qualitative individual and focus group research on migraine [8].

## Recommendations

To optimise migraine treatment, patients must actively participate in condition management [47]. Encouragingly, the current study demonstrates recognition of the self-management pillar. However, the clinical care pathway needs focus. COVID-19 has demonstrated value of telehealth for headache patients [66], and telehealth can be beneficial to reduce neurological waiting list burden [53]. Thus, online consultations may be a viable option to help students with the clinical management pillar, as students are a cohort who are generally au fait with technology [67, 68].

Migraine awareness in the general population could be improved [8] and the current study has highlighted that migraine awareness on university campuses must also be addressed. The imperfect migraine literacy in highly educated students and the external stigma they receive highlight avenues for future migraine awareness campaigns [42]. These campaigns should be initiated on college campuses, and universities should optimise and distribute high-quality, peer-reviewed migraine management information [45]. Other variables that can be potentially improved in a university setting include classroom ergonomic adjustments and providing convenient safe spaces for students to go to during a migraine attack.

Encouragingly, employment law is beginning to recognise migraine as a legitimate medical condition, and some of the attitude changes towards migraine are the result of strong patient advocacy and significant research advances [69]. Interventional studies for migraine management should be conducted on university campuses. They could include such interventions as

exercise trials [70, 71], dietary strategies such as the ketogenic diet [72], and for those students with common migraine-related comorbidities such as mental health disorders, Interpersonal and Social Rhythm Therapy [73]. Further research within the student population could ascertain and highlight the most productive, cost-effective and low-risk treatment strategies for this niche and diverse group of people suffering at a pivotal moment in their lives.

## Conclusions

Optimising migraine treatment in the student cohort will help decrease university absenteeism and presenteeism, increase student productivity, and improve internal and externalised migraine stigma. Improving migraine management will assist in improving overall outcomes, and not only at the college level. If future treatments can address patient requirements, these individuals with migraines can maximise their contribution to the university and society. Thus, studies in this domain have broader relevance.

## Supporting information

**S1 Checklist. PLOS human participants research checklist.**
(DOCX)

**S1 File. SI A1. Recruitment Poster.** Poster used for study recruitment provided via Microsoft Word Document. **SI A2. Participant Information Leaflet.** Leaflet used to describe study provided via Microsoft Word Document. **SI A3. Participant Consent Form.** Informed consent questions prior to visiting electronic link provided via Microsoft Word Document. **SI A4. Migraine Illustration.** Sample of a migraine illustration provided via Microsoft Word Document. **SI A5. Focus Group Questions.** Focus group questions provided prior to study participation, provided via Microsoft Word Document. **SI A6. Standards for Reporting Qualitative Research.** Checklist for study rigor, completed via Microsoft Word Document.
(DOCX)

**S2 File. SI B1. Transcript Data.** All transcript data imported to Microsoft Excel Spreadsheet. **SI B2. Quantification Table.** Grouped data created using Microsoft Excel Spreadsheet. **SI B3. Themes.** Grouped data created using Microsoft Excel Spreadsheet. **SIB4. Additional Information.** Grouped data created using Microsoft Excel Spreadsheet. **SI B5. Reflexivity Memos.** Researcher reflexivity table created using Microsoft Excel Spreadsheet. **SI B6. Study Codebook.** Refined grouped data table created using Microsoft Excel Spreadsheet**. SI B7. Participant Comments.** Participant comments imported to Microsoft Excel Spreadsheet. **SI B Tabs P1-P20. Example Quotes.** Relevant comments used for quotes, imported to Microsoft Excel Spreadsheet.
(XLSX)

## Acknowledgments

The authors thank the students who participated in this research project, providing their time and valuable insights into this debilitating condition.

All authors contributed to the idea, writing, and approval of the work submitted. No other individuals were involved in creating, writing, or approving this manuscript. No other individuals were involved in technical help when compiling this manuscript. All authors discussed the results and commented on the manuscript.

## Author Contributions

**Conceptualization:** Orla Flynn, Catherine Blake, Brona M. Fullen.

**Data curation:** Orla Flynn, Catherine Blake, Brona M. Fullen.

**Formal analysis:** Orla Flynn, Catherine Blake, Brona M. Fullen.

**Investigation:** Orla Flynn, Catherine Blake, Brona M. Fullen.

**Methodology:** Orla Flynn, Catherine Blake, Brona M. Fullen.

**Project administration:** Orla Flynn, Catherine Blake, Brona M. Fullen.

**Supervision:** Orla Flynn, Catherine Blake, Brona M. Fullen.

**Writing – original draft:** Orla Flynn, Catherine Blake, Brona M. Fullen.

**Writing – review & editing:** Orla Flynn, Catherine Blake, Brona M. Fullen.

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
