## [Decision Letter · Decision Letter 0]

5 Apr 2024

PONE-D-24-03960A qualitative exploration of migraine in students attending Irish UniversitiesPLOS ONE

Dear Dr. Flynn,

Thank you for submitting your manuscript to PLOS ONE. After careful consideration, we feel that it has merit but does not fully meet PLOS ONE’s publication criteria as it currently stands. Therefore, we invite you to submit a revised version of the manuscript that addresses the points raised during the review process.

We look forward to receiving your revised manuscript.

Kind regards,

Sercan Ergün

Academic Editor

PLOS ONE

Journal Requirements:

**Additional Editor Comments:**

After careful review by the editorial team and external reviewers, I regret to inform you that your manuscript requires major revisions before it can be considered for publication.

Reviewers' comments:

Reviewer's Responses to Questions

**Comments to the Author**

1. Is the manuscript technically sound, and do the data support the conclusions?

Reviewer #1: Partly

Reviewer #2: Yes

2. Has the statistical analysis been performed appropriately and rigorously? 

Reviewer #1: N/A

Reviewer #2: Yes

3. Have the authors made all data underlying the findings in their manuscript fully available?

Reviewer #1: Yes

Reviewer #2: Yes

4. Is the manuscript presented in an intelligible fashion and written in standard English?

Reviewer #1: Yes

Reviewer #2: Yes

5. Review Comments to the Author

Reviewer #1: Thank you for giving me a chance to review this paper. This is a good study but there are few major comments that should be addressed by authors as listed below:

1. Please clearly write what university is chosen for this study instead of writing “who attended any Irish third level institution”. It seems only one university was chosen for this study. Please clarify why this university was chosen?

2. Please write the study exclusion criteria.

3. It is a qualitative study and not a mixed method study. For the mixed method study you should justify triangulation and how the findings of qualitative and quantitate study support each other but in this study a qualitative design was used with a descriptive analysis of participants characteristics.

4. How did you reach to 20 participants as sample size? Please clarify how did you identify the data saturation is reached?

5. What do you mean of stakeholders in page 6 line 141 where you wrote “convenient for all stakeholders”?

6. It is not clear why and among whom you did focus group and among whom you did in-depth interview? What was the condition for the focus group discussion? Howe and how many participants were chosen for FGD?

7. In what language the interviews were done? If Irish, how about translating after transcribing? Who has done that?

8. How about criteria for trustworthiness? How did you make sure credibility, transferability, Dependability… are achieved?

Reviewer #2: In the abstract: The authors should remove ethical considerations and mention the tools used in the Methods section. In the Results section, they should highlight important findings.

How did the authors diagnose migraine and exclude other types of headaches?

In the Methods, authors should detail their questionnaire sections, including demographic data (age, gender, etc.), medication,..etc

6. PLOS authors have the option to publish the peer review history of their article (what does this mean?). If published, this will include your full peer review and any attached files.

Reviewer #1: No

Reviewer #2: No

---

## [Author Response · Author response to Decision Letter 0]

17 Apr 2024

Sercan Ergün

Academic Editor

PLOS ONE

Re: PONE-D-24-03960

A qualitative exploration of migraine in students attending Irish Universities

Dear Professor Ergün,

Thank you for your feedback and for requesting revisions to our paper. We have implemented the required changes, which are indicated in red in the manuscript and noted here.

Reviewer #1: 

Dear Reviewer #1, thank you for your comments and time spent reviewing our manuscript. We have answered your queries below.

1. Please clearly write what university was chosen for this study instead of "who attended any Irish third-level institution." It seems only one university was chosen for this study. Please clarify why this university was chosen.

The study was open to students attending any third-level institution in Ireland. In the Methods section, we have detailed the recruitment process under the heading ‘Study advertisement’ in red on page 5 of the manuscript.

2. Please write the study exclusion criteria.

The study exclusion criteria were that high school students, students not registered to an Irish third-level institution, or university students without a clinical migraine diagnosis were not permitted to attend. We have modified the 'Methods' section to describe this process. This modification is in red under the heading 'Exclusion criteria’ on page 5 of the manuscript.

3. It is a qualitative study, not a mixed-method study. For a mixed-method study, you should justify triangulation and explain how the findings of the qualitative and quantitative studies support each other, but in this study, a qualitative design was used with a descriptive analysis of the participants' characteristics.

We note that the term mixed methods was used in error, and we have modified the sentence to state that this is a qualitative study under the heading ‘Research Design’ on page 7 of the manuscript.

4. How did you reach 20 participants as a sample size? Please clarify how you identified that the data saturation was reached.

We calculated the sample size based on qualitative individual and focus group literature. We considered saturation under the remit of information power. In the 'Methods' section of the manuscript, we have detailed the sample size process in red under the heading 'Sample Size’ and explained why we ceased data collection at 20 participants on pages 6 and 7 of the manuscript.

5. What do you mean by stakeholders on page 6, line 141, where you wrote "convenient for all stakeholders"?

We used the term stakeholders in error. We have removed this sentence.

6. It is not clear why and among whom you conducted the focus group and in-depth interview. What was the condition for the focus group discussion? How and how many participants were chosen for FGD?

We have provided the rationale for individual or focus groups, indicated in red under the heading 'Sample Size' on page 6 of the manuscript. We have outlined the rationale for the focus group size, indicated in red under the heading 'Timeframe' on page 7 of the manuscript. We have also described who participated in the individual and focus groups within the manuscript, marked in red under the heading 'Results’ on page 14 of the manuscript.

7. In what language the interviews were done? If Irish, how about translating after transcribing? Who has done that?

The interviews were conducted in English. We have added this under the heading Language on page 7 of the manuscript. 

8. How about criteria for trustworthiness? How did you make sure credibility, transferability, Dependability… are achieved?

We have revised the 'Methods' section under 'Qualitative data analysis' on page 12 of the manuscript and 'Research team, reflexivity, and trustworthiness on page 13 of the manuscript. We have outlined our meticulous and detailed data collection and analysis approach in red in these sections. 

We have also provided further information in additional Supporting Information (SI) B. This spreadsheet allows for the reproducibility of the data extraction and analysis and thus demonstrates the credibility, transferability, and dependability of the results. This spreadsheet has been provided within the revised supplementary content. It is indicated in red in the supplementary content table and referred to in red in the manuscript's revised 'Methods' section on pages 12 and 13 of the manuscript. 

Reviewer #2

Dear Reviewer #2, thank you for your comments and time spent reviewing our manuscript.

We have answered your queries in red below.

Reviewer #2: In the abstract, The authors should remove ethical considerations and mention the tools used in the Methods section. In the Results section, they should highlight important findings.

The ethical considerations have been removed from the abstract.

The tools used have been detailed in the methods section in red in the abstract.

The important findings have been highlighted in red in the abstract.

How did the authors diagnose migraine and exclude other types of headaches?

The authors did not diagnose migraine, but the study inclusion criteria were that potential participants must have been clinically diagnosed with migraine (via doctor or neurologist) to participate. We have modified the ‘Methods’ section under the heading ‘Exclusion criteria’ which is indicated in red on page 5 of the manuscript. 

In the Methods, authors should detail their questionnaire sections, including demographic data (age, gender, etc.), medication,..etc

The ‘Methods’ section has been revised to include an additional table, ‘Table 1: Demographics Questions’. This change is indicated in red under the ‘Methods’ section on pages 8 and 9 of the manuscript.

Additional Requirements:

1 A marked-up copy of your manuscript highlighting changes made to the original version. You should upload this as a separate file labeled 'Revised Manuscript with Track Changes .'An unmarked version of your revised paper without tracked changes. You should upload this as a separate file labeled 'Manuscript.'

Two versions of the manuscript have been uploaded; 1.Revised Manuscript with Track Changes, and 2. Manuscript'

2. Guidelines for resubmitting your figure files are available below the reviewer comments at the end of this letter. While revising your submission, please upload your figure files to the Preflight Analysis and Conversion Engine (PACE) digital diagnostic tool, https://pacev2.apexcovantage.com/. PACE helps ensure that figures meet PLOS requirements. 

Figure 1 was uploaded to the PACE diagnostic tool and modified to fit the Plos One requirements. It has been reuploaded in Tif format. 

3. Please include your full ethics statement in the 'Methods' section of your manuscript file. In your statement, please include the full name of the IRB or ethics committee who approved or waived your study, as well as whether or not you obtained informed written or verbal consent. If consent was waived for your study, please include this information in your statement as well.

The full ethics statement has been included under the heading ‘Ethical approval & informed consent’ and is indicated in red in the ‘Methods’ section on page 4 of the manuscript.

The manuscript meets PLOS ONE's style requirements, including file naming requirements. 

2. PLOS requires an ORCID iD for the corresponding author in Editorial Manager on papers submitted after December 6th, 2016. Please ensure that you have an ORCID iD and that it is validated in Editorial Manager. To do this, go to 'Update my Information' (in the upper left-hand corner of the main menu) and click on the Fetch/Validate link next to the ORCID field. This will take you to the ORCID site and allow you to create a new ID or authenticate a pre-existing iD in Editorial Manager. 

The ORCID ID for the corresponding author has been validated in Editorial Manager. 

Kind regards,

Ms. Orla Flynn, PhD Candidate, University College Dublin

---

## [Decision Letter · Decision Letter 1]

3 Jun 2024

A qualitative exploration of migraine in students attending Irish Universities

PONE-D-24-03960R1

Dear Dr. Flynn,

We’re pleased to inform you that your manuscript has been judged scientifically suitable for publication and will be formally accepted for publication once it meets all outstanding technical requirements.

Kind regards,

Sercan Ergün

Academic Editor

PLOS ONE

Additional Editor Comments (optional):

The manuscript is suitable for the publication in this form.

Reviewers' comments:

Reviewer's Responses to Questions

**Comments to the Author**

1. If the authors have adequately addressed your comments raised in a previous round of review and you feel that this manuscript is now acceptable for publication, you may indicate that here to bypass the “Comments to the Author” section, enter your conflict of interest statement in the “Confidential to Editor” section, and submit your "Accept" recommendation.

Reviewer #1: All comments have been addressed

Reviewer #3: All comments have been addressed

2. Is the manuscript technically sound, and do the data support the conclusions?

Reviewer #1: (No Response)

Reviewer #3: Yes

3. Has the statistical analysis been performed appropriately and rigorously? 

Reviewer #1: N/A

Reviewer #3: Yes

4. Have the authors made all data underlying the findings in their manuscript fully available?

Reviewer #1: Yes

Reviewer #3: Yes

5. Is the manuscript presented in an intelligible fashion and written in standard English?

Reviewer #1: Yes

Reviewer #3: Yes

6. Review Comments to the Author

Reviewer #1: Thank you for addressing all the comments and there is no new comment. Your paper is much promoted.

Reviewer #3: The article titled "A qualitative exploration of migraine in students attending Irish Universities" is accepted.

The revision has made well following suggestions.

7. PLOS authors have the option to publish the peer review history of their article (what does this mean?). If published, this will include your full peer review and any attached files.

Reviewer #1: **Yes: **Masoud Mohammadnezhad

Reviewer #3: No

---

## [Editor Report · Acceptance letter]

13 Jun 2024

PONE-D-24-03960R1 

PLOS ONE

Dear Dr. Flynn, 

I'm pleased to inform you that your manuscript has been deemed suitable for publication in PLOS ONE. Congratulations! Your manuscript is now being handed over to our production team.

Kind regards, 

on behalf of

Dr. Sercan Ergün 

Academic Editor

PLOS ONE